# Neutralizing human monoclonal antibodies to poliovirus map to the receptor binding site

Benjamin T. Waddey[1,2], Andrew J. Charnesky[3], Julia E. Faust [3], Nadia M. DiNunno[4], Rama Devudu Puligedda[5], Sung Hyun Cho [6], Carol M. Bator[2], Steven D. Dong[7], Kutub Mahmood[7], Konstantin M. Chumakov[8], Scott K. Dessain[5] & Susan L. Hafenstein [1,2,9] ✉

Poliovirus remains a serious threat to human health. Complete eradication of wild-type poliovirus has not yet succeeded, making the development of successful antivirals critical. Microneutralization assays against all three poliovirus serotypes identified a panel of human monoclonal IgGs, which are either serotype-specific or cross-neutralizing. Here, through cryoEM single particle analysis, we solved high resolution structures of four distinct poliovirus-FAb complexes. These antibodies bind to capsids at the circular depression (canyon) surrounding the icosahedral five-fold symmetry axis, which is also the binding site of the poliovirus receptor (PVR). Analysis of these structures confirms overlap of FAb contacts on the viral capsid with those of PVR. For three of the FABs, the capsid residues are identified that dictate serotype-specific recognition. Contacts for the cross-neutralizing mAb 10D2 are located deep in the capsid canyon. These structural analyses indicate that antibody competition with the receptor likely leads to neutralization of virus particles and inhibition of poliovirus entry into host cells. Thus, the human IgGs studied here may facilitate development of therapeutics for the ongoing efforts in global eradication of poliovirus.

Poliovirus (PV) eradication efforts have been implemented for several decades, with the first vaccination programs beginning in the 1950s[1,2]. Poliovirus route of infection is through pharyngeal and fecal–oral transmission, with most cases presenting as asymptomatic. Gastrointestinal symptoms occur in ~5% of cases, with the severe clinical outcome of acute flaccid paralysis occurring in ~1% of cases[3]. There are three serotypes of poliovirus, distinguished by differences in genome sequences and conformational epitopes[4]. Exposure to one serotype of

PV does not protect against other serotypes, and current vaccination programs inoculate against all three serotypes of PV. Vaccination efforts have had a profound effect on the eradication of wild-type poliovirus, with serotypes 2 and 3 declared eradicated, though serotype 1 remains endemic to Afghanistan and Pakistan[5]. Complete eradication of poliovirus has been stymied by vaccine hesitancy and the necessary continued use of the live attenuated vaccine. Although stable and easy to administer, the Sabin attenuated virus in the oral PV

[1]College of Biological Sciences, University of Minnesota, Twin Cities, St. Paul, MN, USA. [2]The Hormel Institute, University of Minnesota, Austin, MN, USA. [3]Eberly College of Science, The Pennsylvania State University, State College, Hershey, PA, USA. [4]College of Medicine, The Pennsylvania State University, Hershey, PA, USA. [5]Lankenau Institute for Medical Research, Wynnewood, PA, USA. [6]The Huck Institutes of Life Science, The Pennsylvania State University, State College, Pennsylvania, PA, USA. [7]Center for Vaccine Innovation and Access, PATH, Seattle, WA, USA. [8]Department of Microbiology, Immunology, and Tropical Diseases, George Washington University, Washington, DC, USA. [9]Mayo Clinic, Department of Infectious Diseases, Rochester, MN, USA. ✉e-mail: hafen@umn.edu

vaccine can revert to a virulent state, resulting in vaccine-derived polioviruses (VDPVs)[6–8]. Circulating VDPVs (cVDPV) of all three serotypes are present globally and have the potential to spread throughout under-immunized communities, including areas where wild-type polio transmission has been effectively eradicated previously[8,9]. In addition to cVDPV, derivatives of Sabin strains used in OPV can establish chronic infection in individuals with primary immunodeficiency[10]. Thus, the development of new antivirals is critically necessary to support efforts for the eradication of poliovirus.

Poliovirus is a positive sense ssRNA virus in the family *Picornaviridae*. The viral capsid is ~30 nm in diameter and comprises four structural proteins (VP1–4) arranged with $T = 1$ (pseudo-$T = 3$) icosahedral symmetry. Structural features of these viruses include a raised plateau termed the mesa at the five-fold symmetry axis, which is surrounded by a depression known as the canyon[11]. The canyon is the site of poliovirus receptor (PVR) binding[12–14]. At the bottom of the canyon is an opening to a hydrophobic pocket that contains a lipid molecule, or pocket factor (PF), which stabilizes the capsid. Upon PVR binding, this lipid exits the pocket, allowing conformational changes to occur that result in the formation of the altered particle, or A-Particle. The A-Particle is a true entry intermediate, which is characterized by the externalization of VP4 and the N-termini of VP1 from the interior of the capsid. These conformational changes further facilitate particle entry and genome release within the host cell[15–19]. Since the receptor-binding site is conserved across all three serotypes of poliovirus and is essential for virus entry, this site might be exploited for the development of biologics.

High-resolution 3D structures of poliovirus, both alone and with receptor or antibody fragments bound, have given insight into receptor contacts and neutralizing epitopes on the capsid surface[14,20–24]. The neutralizing epitopes of poliovirus were first determined using escape mutant analysis in mice[25–27] and cryoEM structures have previously characterized neutralizing epitopes of poliovirus by mapping both human–chimpanzee chimeric antibodies and camelid nanobodies to the PV capsid[24,28]. Recently, high-resolution cryoEM structures from our group demonstrated that 9H2, a human monoclonal antibody neutralizing all three PV serotypes, had a distinct epitope compared to those of established murine models[23].

Patient-derived, human monoclonal IgGs have been previously isolated as potential biologic candidates. Five of these mAbs (9H2, 10D2, 5E12, 2E1, and 6B5) have been shown to neutralize poliovirus in vitro. Both 9H2 and 10D2 are cross-neutralizing, neutralizing more than one PV serotype, whereas the others are serotype-specific[23,29]. Previous analysis of 9H2 showed that this antibody binds around the five-fold icosahedral symmetry axis of the PV capsid, overlapping the PVR-binding site, outcompetes soluble PVR for capsid binding, and for PV1 causes expulsion of PF. Here, we have solved the high-resolution structures of poliovirus with fragments antigen-binding (FAbs) of the neutralizing mAbs 10D2, 5E12, 2E1, and 6B5. Through structural analyses, we have determined that all four mAbs bind around the fivefold icosahedral symmetry axis similarly to 9H2; no major conformational changes were induced by FAb binding, and pocket factors are retained in each complex. We were able to map each FAb to the capsid to identify conformational epitopes and the footprints. We determined serotype-specific interactions of these antibodies and showed that contacts overlap with both mAb 9H2 and the poliovirus receptor. This investigation adds considerable information to what residues trigger PF release, induce A-particle formation, and dictate serotype-specificity. Thus, these specific results may aid us significantly in advancing the development of biologicals against PV.

## Results
### Five-fold binding FAbs saturate PV capsid surface
To investigate the conformational epitopes of these antibodies by cryoEM, fragment antigen binding (FAb) was generated by papain digestion and purified. These FAbs were incubated in excess (3 FAb: 1

virus asymmetric unit) with either Sabin inactivated PV1 (5E12), wild-type PV2 (10D2) or Sabin-inactivated PV3 (2E1 and 6B5) for 30 min at room temperature. Resulting complexes were screened by negative stain TEM and vitrified on continuous carbon-coated Quantifoil grids prior to cryoEM data collection. Data were collected using either a Talos Arctica 200 kV microscope at the Penn State cryoEM facility or an FEI Titan Krios G2i microscope at the University of Minnesota Hormel Institute. Micrographs of all complexes showed virions with densities protruding from the capsid surface, consistent with bound Fab and central capsid densities that were consistent with the packaged genome (Fig. S1a).

### Reconstruction and model refinement of PV−FAb complexes
For each PV–FAb complex, several rounds of particle picking and 2D classification were used to select the best 2D classes for homogeneous 3D reconstruction with icosahedral symmetry imposed. After reconstruction and refinement, the resolution of the complex maps ranged from ~2.8 to 3.7 Å. For each complex map, the density magnitude of FAb was equal to that of the capsid as seen in the central section (Fig. 1), which also showed amorphous genomic density. The 3D map for each complex showed a similar binding mode with all 60 symmetry-related copies of the FAb bound to the capsid around the fivefold axis without apparent steric hindrance (Fig. 1a). Local resolution mapping of each complex showed poorer resolution at the hinge region of the FAb, likely due to the inherent flexibility of this structure (Fig. S1D). Overall, local resolution showed strong, even resolution throughout the capsid and no regions of poor resolution induced by FAb binding. Thus, the complex maps were of sufficient resolution and quality for modeling both the virus capsid and the FAb variable domain. For the capsid, the build was initiated using the previously published complex structures of FAb 9H2 and poliovirus (PDB ID 8E8Z, 8E8S, and 8E8X for SIPV1, PV2, and SIPV3-binding complexes, respectively)[23]. Protein models for the FAb variable domains were generated by submitting the sequence to SabPred AbodyBuilder2[30]. The build for each FAb was initiated with the model placed into the variable domain and adjusted throughout all residues by fit. This initial placement enabled building for all three complementarity-determining regions (CDRs) for both the heavy and light chains for all four mAbs, along with the entirety of each framing region. For all complex maps, capsid proteins VP1–VP4 were also built successfully and included residues VP1 22–302, VP2 9–273, VP3 1–238, and VP4 residues 2–15 and 24–68.

### No gross conformational changes were observed upon FAb binding
Possible conformational changes imposed by FAb binding to viral capsids were evaluated by superimposing each of our capsid models with the corresponding PV capsid crystal structures (PDB ID 1HXS, 1EAH, and 1PVC for PV1, PV2, and Sabin PV3, respectively)[31–33]. The α-carbon root mean squared deviation (RMSD) for the FAb–virus complexes ranged from ~0.5 to 1.1 Å compared to their representative crystal structures, constituting a lack of global conformational change across each capsid. Possible conformational changes of the capsid were also investigated by comparing the complexes of Fab-PV capsids presented here and previous virus complexes with cross-neutralizing antibody 9H2[23]. The α-carbon root mean squared deviation (RMSD) for the FAb–virus complexes once again ranged from ~0.5 to 1.1 Å compared to their 9H2-PV counterparts (Table S2). While no gross conformational changes were observed, local conformational changes within the FAb interface are described, ranging from ~3 to 7 Å RMSD per residue compared to representative crystal structures (Fig. S3). Common sites of conformational changes include the VP1 BC loop 95–103 and VP1 GH loop 230–237, which are also common sites of conformational changes induced by poliovirus receptor binding. The GH loop of VP1 is in the doorstop "up" position for each complex compared to their reference crystal structures, a conformation that is

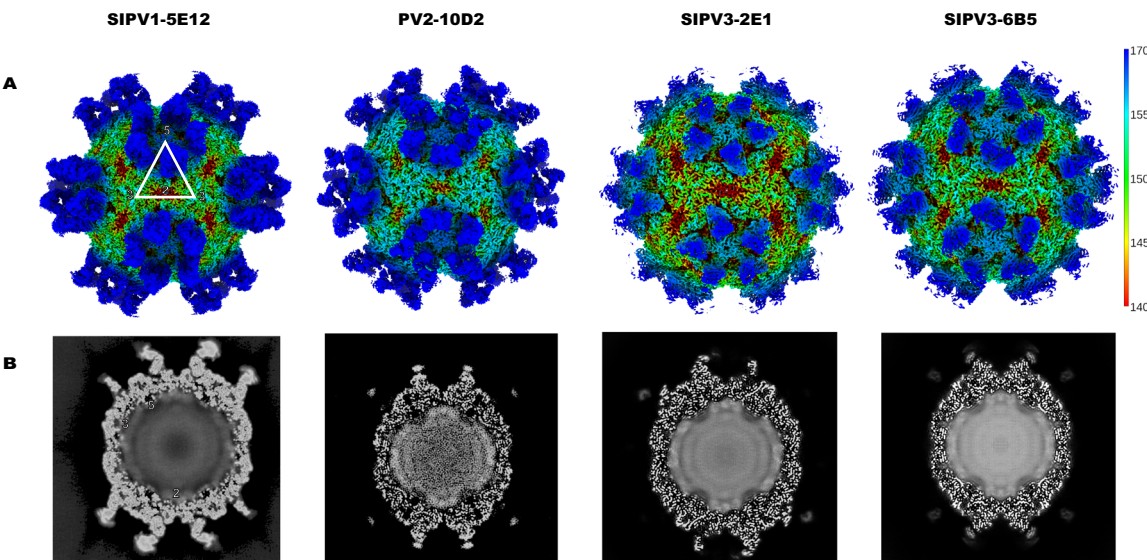

**Fig. 1 | cryoEM of PV–FAb complexes. A** Surface rendered map of PV–FAb complexes, colored by radius from the center of the map in angstroms (color key shown). Icosahedral five-fold, three-fold, and two-fold symmetry axes of the capsid surface are labeled for orientation (white). **B** Central section of EM maps for corresponding complexes, showing FAb fragments at the five-fold axis of each PV capsid, equal in density magnitude to that of the capsid. Icosahedral five-fold, three-fold, and two-fold symmetry axes of the capsid cross-section are labeled for orientation.

present in both receptor-bound and A-particle conformations of PV capsids[14,17] (Fig. S4).

Density for the pocket factor molecule, in most cases equal in magnitude to that of the viral capsid, is present in all virus–FAb complexes shown (Fig. S5). This corroborates previous findings of pocket factor density observed in complexes formed with cross-neutralizing antibody 9H2 to both PV2 and PV3 capsids, but not PV1[23]. Interestingly, here strong pocket factor density for the SIPV1–5E12 complex is also observed, equal in density to that of the capsid. These findings, coupled with the presence of genome density and density for the interior capsid protein VP4 within the EM map, confirm that FAb binding does not trigger conformational changes leading to the formation of the Altered-particle entry intermediate.

### Capsid contacts for each antibody overlap with PVR and 9H2 binding sites

FAb–virus contacts were defined as amino acid residues on the capsid within 0.4 Van der Waal's radii of FAb residues[34]. All FAbs bind near the five-fold icosahedral symmetry axis of the PV capsid, with contacts mapping into and near the canyon region (Fig. 2, Table 1A). Each FAb contacts VP1, with additional contacts mapping to VP2 of the same protomer in all complexes except for the SIPV3–2E1 complex. Contacts on the VP3 chain of the neighboring protomer were seen in all complexes except for the PV2–10D2 complex. These contacts with the virus surface make the footprints of 6B5, 2E1, and 10D2 highly similar. However, the footprint of 5E12 was unique with respect to contacts located on the VP1 chain of the neighboring protomer (Fig. 2).

Each mAb has contacts on the virus capsid that overlap with those of the poliovirus receptor, ranging from 3 to 13 total common contacts per mAb (Figs. 3, S2 and Table 2). Most common contacts map to VP1, with a few contacts on either VP2 or VP3 that overlapped for any complex. Of the four FAbs, only two (6B5 and 5E12) shared overlapping contacts with all three exterior capsid proteins VP1–3 in their footprints, whereas the 2E1 footprint shared overlaps with VP1, VP3, and 10D2, which only shared common contacts with VP1 (Table 2).

Each mAb also has several contacts shared with 9H2, the previously characterized pan-neutralizing mAb that also binds at the five-fold symmetry axis on the PV capsid. The total number of shared contacts between 9H2 and each mAb ranges from 5 to 9 contacts, with most of these on capsid protein VP1. MAb 10D2 has significant overlap with 9H2 compared to the other mAbs shown, with 8 of its 10 total capsid contacts overlapping directly with 9H2's PV2 footprint (Fig. S7, Table S3).

Regarding the paratope of each mAb on the PV capsid surface, each FAb protein had variable numbers of residues binding to the viral capsid, ranging from 6 to 16 total residues per complex (Table 1B). Most of these contacts were from the FAb heavy chain, with a small portion of contacts (2–7 total) from the light chain. Most contacts were contributed by the complementary-determining regions (CDRs) of the FAb variable domain, namely CDR3 for both the heavy and light chains, with contacts from the FAb-framing regions only present for two of the PV–FAb complexes (Table S4).

### Contacts conferring serotype-specific neutralization of five-fold-binding mAbs

Some of the mAbs presented have "sequence-equivalent" contacts, those at the same amino acid position in the capsid proteins when the three serotype sequences are aligned (Fig. S13). The comparison of these contacts indicates which interactions drive serotype-specific binding and neutralization. The mAb 5E12 specifically neutralizes only PV1, whereas 6B5 and 2E1 neutralize only PV3. While there are differences in identity of amino acid residues at some positions, two specific residues at positions 100 and 168 were considered as being critical to serotype-specific neutralization of these mAbs, due to the lack of conservation between each of the three PV serotypes.

5E12 and 2E1 both contact the residue at VP1 position 168. In SPV1, this residue is a glutamate and is predicted to make hydrogen bonds and electrostatic interactions with residue R100 in CDR3 of the 5E12 heavy chain. Conversely, in PV3, this residue is a lysine, leading to electrostatic repulsion and subsequent lack of binding at this residue position. This lack of contact is also true for PV2, where there is a glycine residue that is unable to form a predicted H-bond with R100. Conversely, 2E1 makes electrostatic interactions with VP1 K168 through residue E56 in its light chain, whereas repulsive charge contacts at E168 of SPV1 lead to the lack of contact at this position (Fig. 4A and B).

Another important site is found at position 100 of VP1. Both 5E12 and 6B5 contact this residue, an asparagine in type 1 poliovirus and arginine in both type 2 and type 3 poliovirus. For 5E12, predicted hydrogen bonds between the terminal amine at residue N1 of the heavy

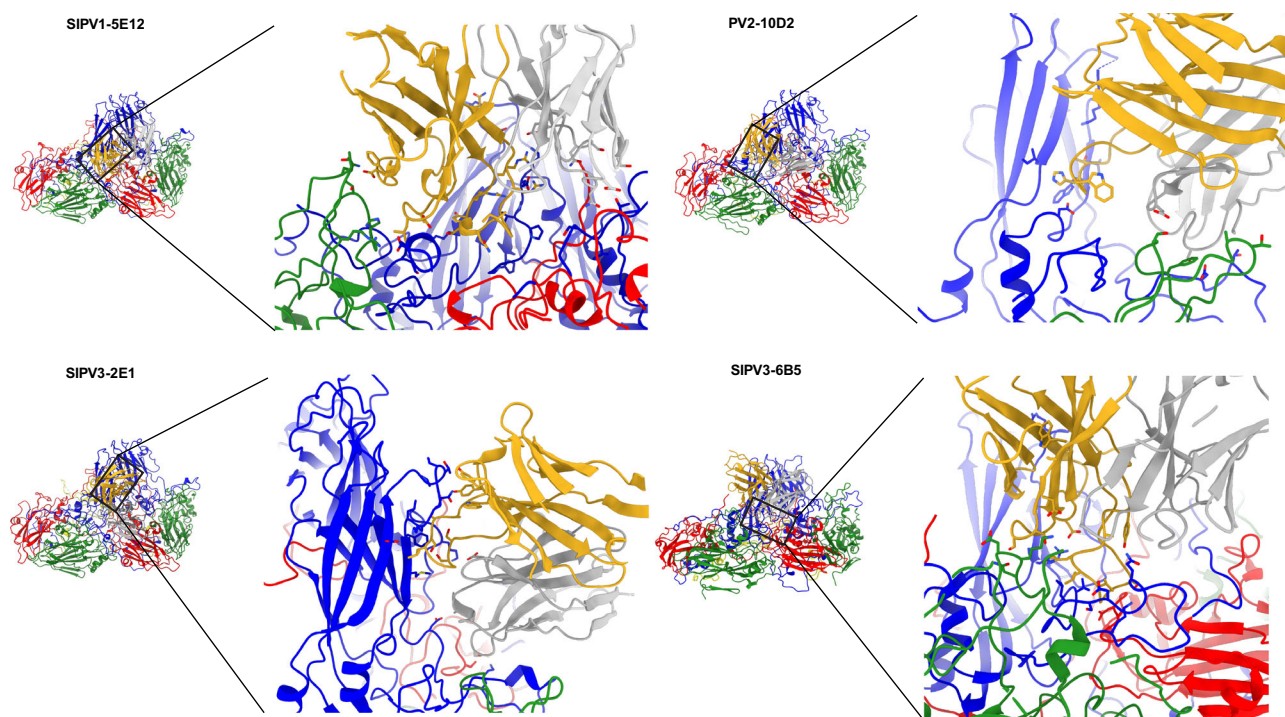

**Fig. 2 | FAb and virus contacts in the binding interface.** Capsid proteins VP1–4 are colored blue, green, red, and yellow, respectively. Heavy and light chains of each FAb are colored goldenrod and dark gray, respectively. *Left* View is shown as the interface of the Fab binding that includes two adjacent capsid protomers. Binding interface is outlined in a black box. *Right* Zoomed inset of FAb-capsid binding interface, shown as sticks with additional coloring by heteroatom (no hydrogens are shown).

## Table 1 | FAb contacts on PV capsid

| A | | | |
|---|---|---|---|
| **FAb** | **VP1 residues** | **VP2 residues** | **VP3 residues** |
| 5E12 | N100, D102, K109, P161, P162, G163, A164, P165, V166, P167, E168, K169, D226, S227, L228, G289, T292, L294, S295 | T138, H142, Q167, T168, R172 | K62, T179, I180, D181, K234 |
| 10D2 | I89, K103, V107, W108, D114, T223, D226 | T137, F141, A166, T167 | N/A |
| 2E1 | E91, P97, T98, T99, Q102, K103, L104, F105, M107, K168, K214, D236 | N/A | S183 |
| 6B5 | T99, R100, R109, K214, Q224, D227, S228, L229, V235 | D137, K138, Q139 | D182, S183 |
| B | | | |
| **FAb** | **Heavy chain residues** | | **Light chain residues** |
| 5E12 | Q1, D54, L56, F71, D73, R100, R102, Y103, S105, N106, I107, R111 | | D28, N36, S65, N66, E68, T69, S83 |
| 10D2 | T108, H109, G110, W111 | | S27, D113 |
| 2E1 | Y32, W33, Y52, A103, S104, S105, L106, S109, S110 | | A35, E56, S83 |
| 6B5 | E23, S25, D31, S56, S57, D59, S105, G106, G107, V108, L109, H110, Y112 | | Y36, S114 |

(A) VP1–3 residues that are part of each FAb binding footprint. Amino acid residues labeled by single-letter code and numbered according to sequence position (N/A: not applicable). Virus residues are reported as sequence equivalent per an alignment to SPV1. (B) Paratope residues of FAb binding to each virus capsid, labeled by single letter code and numbered by sequence position.

chain and the carbonyl in the N100 side chain mediate contact at this residue, whereas the side chain of the arginine residue for PV3 would sterically clash with the n-terminal amine of the heavy chain variable region, preventing binding. In the case of 6B5 binding to residue VP1 R100, the terminal carbonyl groups of D23 in the 6B5 heavy chain make electrostatic contacts with R100, whereas the N100 side chain in SPV1 is out of position to contact D23 (Fig. 4C and D).

### Contacts in the canyon floor confer cross-serotype neutralization of poliovirus in mAB 10D2

The 10D2 mAb described here neutralizes serotypes 2 and 3, whereas the 9H2 mAb analyzed previously neutralizes all three serotypes of PV. Antibody 9H2 recognizes capsids via a CDR loop that binds into the canyon, where residue S128 in 9H2's heavy chain is predicted to make hydrogen bonds with residues D114 and W108, adjacent to the pocket factor molecule[23]. The CDR loop 3 of mAb 10D2 also binds into the canyon floor, with residue G110 contacting residue D114 of VP1. Additionally, the R-group terminal alcohol in residue T108 of the 10D2 CDR3 makes a predicted hydrogen bond with V107's carboxyl group and overlaps with residue W108 on VP1. Each of these contacts, along with contacts in the VP1 GH loop at residues 223 and 226, helps stabilize the CDR loop 3 in the canyon floor (Fig. 5).

### Five-fold binding mAbs neutralize as FAb fragments

The distance of the heavy chain C-termini of each FAb ranges from 59 to 75 Å, with symmetry-related FAb copies angled away from one

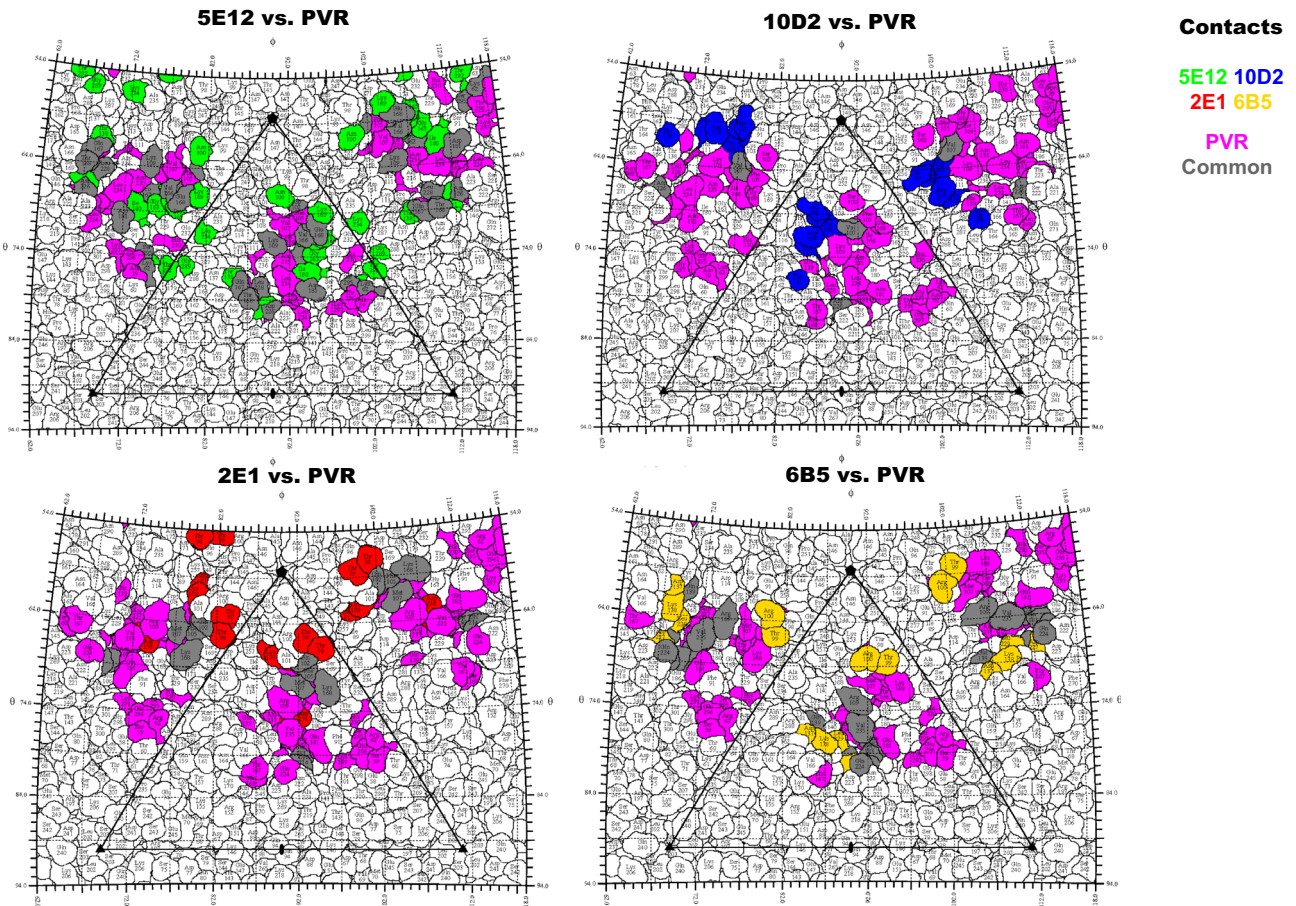

**Fig. 3 | Five-fold binding FAbs overlap PVR-binding footprint.** Road maps of the virus surface, shown as a projection of the icosahedral asymmetric unit indicated by the triangular boundary, with polar angles φ and θ representing the latitude and longitude of a point on the viral surface, respectively[53]. Residues corresponding to either the FAb footprints alone, the PVR footprint alone, or common contacts between both footprints are shaded as indicated in color key.

## Table 2 | Common contacts between FAb and PVR footprint on the capsid

| | 5E12 | Common | PVR |
|---|---|---|---|
| VP1 | 100,161,163,164,165,169,227,289,292,294 | 102,109,162,166, 167,168,226,228, 295 | 105,106,107,205, 214,224,234,235, 237,239,296,297 |
| VP2 | 138,172 | 142,167,168 | 139 |
| VP3 | 179,180,234 | 62,181 | 59,176,178,182, 183, 184, 230 |

| | 10D2 | Common | PVR |
|---|---|---|---|
| VP1 | 89, 103, 108, 114, 223 | 107,226 | 102,105, 106,109,162, 166,167,168,205, 214,224,228,234, 235,237,239,295, 296,297 |
| VP2 | 137,141, 166 | 167 | 139,142, 168 |
| VP3 | N/A | N/A | 59,62,176,178,181,182,183,184,230 |

| | 2E1 | Common | PVR |
|---|---|---|---|
| VP1 | 91,97,98,99,103, 104,236 | 102,105,107,168, 214 | 106,109,162,166, 167,205,224,226, 228,234,235,237, 239,295,296,297 |
| VP2 | N/A | N/A | 139,142,167,168 |
| VP3 | N/A | N/A | 59,62,176,178,181,182,184, 230 |
| | 6B5 | Common | PVR |
| VP1 | 99,100, 227, 229 | 109, 214, 224, 228, 235 | 102,105,106,107, 162,166,167,168, 205, 226, 234,237, 239,295,296,297 |
| VP2 | 137, 138 | 139 | 142, 167, 168 |
| VP3 | N/A | 182, 183 | 59,62,176,178,181,184,230 |

Amino acid residues are numbered per sequence alignment to SPV1 for the position of each capsid protein.
N/A: not applicable.

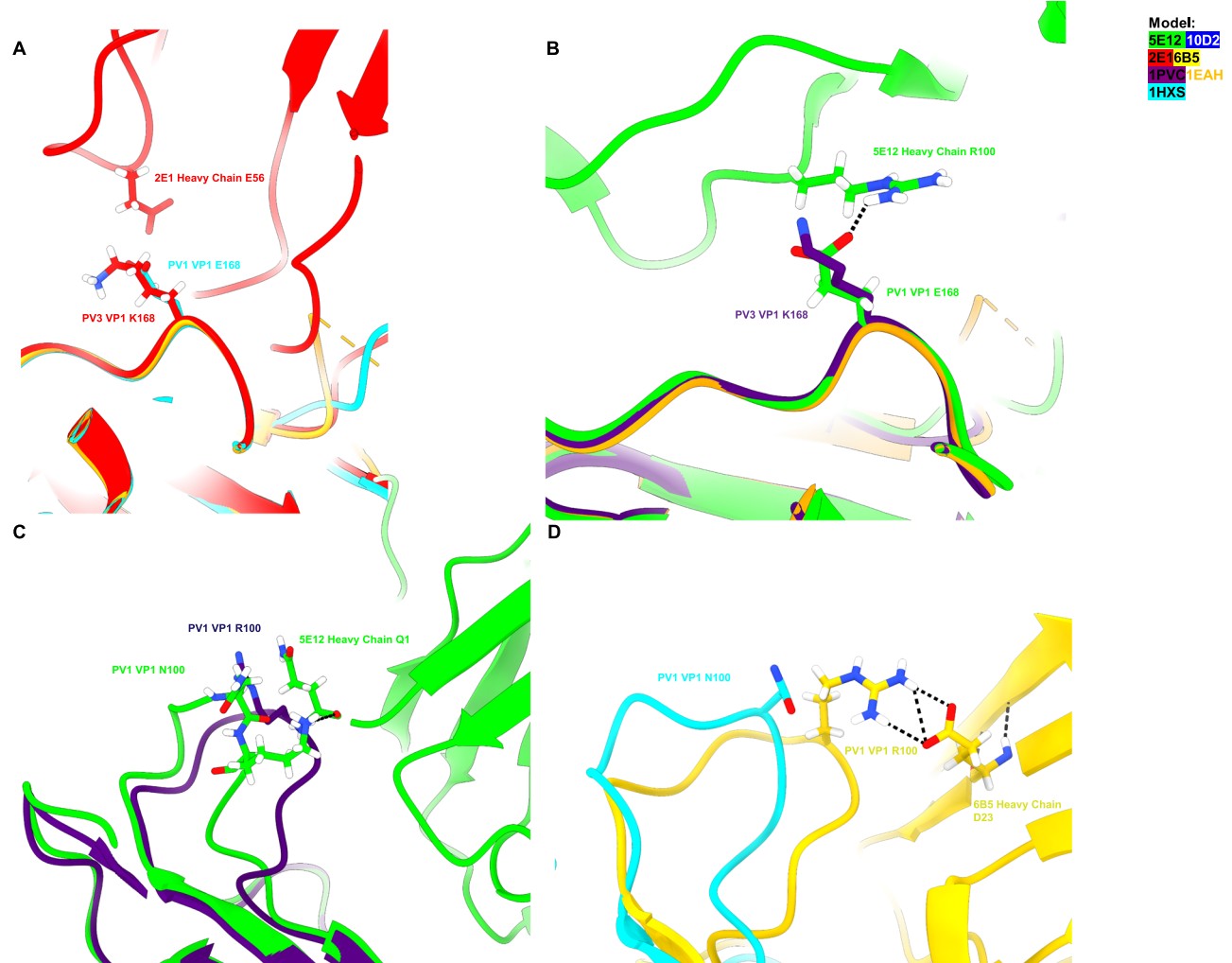

**Fig. 4 | VP1 residues 168 and 100 confer serotype-specific neutralization for five-fold binding FAbs.** Apo PV crystal structures (PDB ID 1HXS, 1EAH, and 1PVC)[30–32] and FAb-bound PV models shown in this study are colored according to key, with additional coloring for residues by heteroatom and hydrogen bonds represented as black dashed lines. **A** and **B** 2E1 heavy chain residue E23 makes predicted electrostatic contact with VP1 residue K168 in PV3, whereas in PV1, there would be an electrostatic repulsion by residue E168 in PV1 and a lack of hydrogen bonding. The inverse is indicated for 5E12 heavy chain R100, where E168 in SPV1 can make electrostatic and H-bonding contact with R100; however, binding is abrogated with K168 in SPV3. **C** and **D** Residue Q1 of the 5E12 heavy chain makes H-bond contacts with VP1 N100 of SPV1, whereas in SPV3, the R100 would sterically clash with the Q1 residues. Conversely, residue D23 of 6B5 heavy chain makes electrostatic and H-bond contact with R100 of SPV3, whereas N100 of SPV1 would be out of position for this contact.

another, tangential to the capsid. This binding mode, coupled with the binding angles relative to the capsid, would prevent bivalent binding and accommodate monovalent binding to the capsid[35,36]. The mode of binding is similar to the monovalent binding of 9H2 FAb to PV capsids, which was sufficient to neutralize serotypes 1 and 3 of poliovirus in-vitro[23].

Neutralization efficacy of each of the mAbs presented here towards either SPV1 or SPV3 was assessed by microneutralization assays with the full-length mAb, with effective neutralization concentrations ranging from $2.0*10^{-2}$ to $2.0*10^{-6}$ mg/mL. We performed further microneutralization assays with 10-fold serial dilutions of FAb fragments towards either SPV1 (for 5E12) and SPV3 (for all other mAbs). All four antibodies neutralized the virus as FAb, with effective neutralizing concentrations ranging from $2.0*10^{-2}$ to $2.0*10^{-5}$ mg/mL (Fig. S14).

## Discussion

Here, we describe the structural work that informs the mechanism of neutralization by four distinct mAbs. These mAbs have a common binding mode at the five-fold symmetry axis of PV capsids and the ability to neutralize both as full-length mAb and FAb. Each of these

mAbs shares capsid contacts with poliovirus receptor on the PV capsid, suggesting that the main mechanism of neutralization is competition for PVR binding, leading to loss of A-particle formation and defective entry.

Antigenic sites N-AgI-III for poliovirus were established by analyzing escape mutant viruses towards murine monoclonal antibodies[25–27]. Each mAb shown here has little overlap with any murine antigenic site, with as few as a single overlapping contact per mAb for 10D2. 5E12 displays the most overlap with five total contacts, including contacts at each of the three characterized sites (VP1 100 for Ag-I, VP1 226 and VP2 167-168 for N-AgII, and VP1 289 for N-AgIII), followed by 2E1 (VP1 91, 97-99 in N-AgI) and 6B5 (VP1 99-100 in N-AgI and VP1 224 in N-AgII). Only one of 10D2's 10 capsid contacts overlaps with any murine antigenic site, at VP1 226 (representing N-AgII). These data, in accordance with previous analysis of mAb 9H2, reveal that human epitopes towards PV are distinct from those previously established using murine models, which may prove important for the development of biologics and stable vaccines.

We have identified several residues on the PV capsid that are critical for serotype-dependent neutralization of mAbs 5E12, 2E1, and 6B5

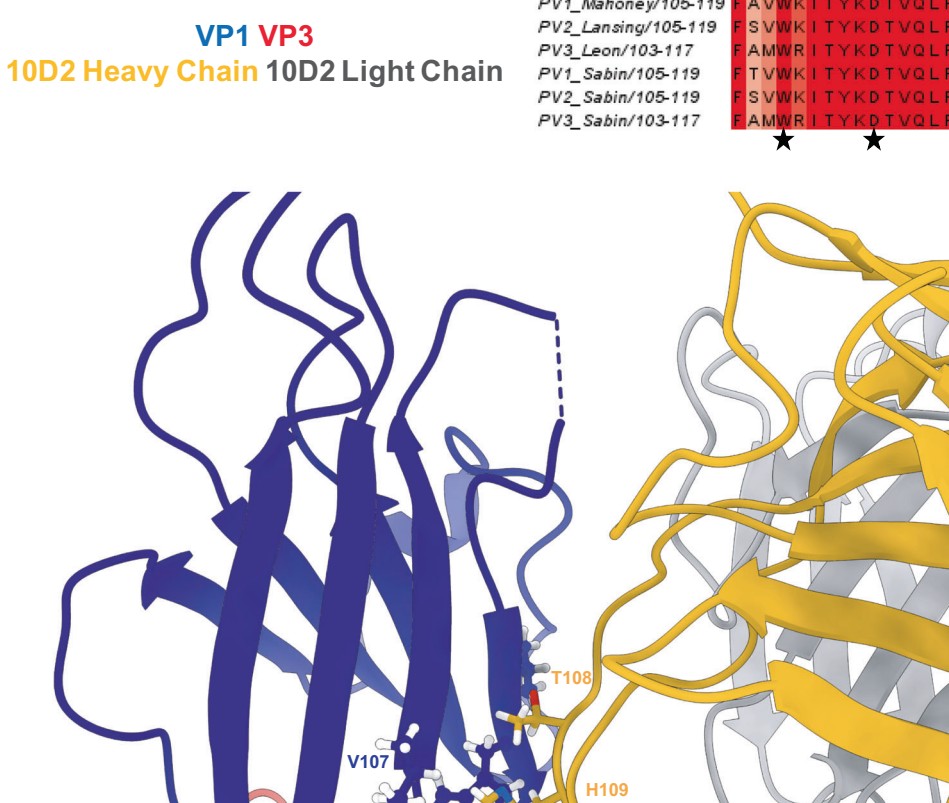

**Fig. 5 | 10D2's CDR loop likely confers cross-serotype neutralization.** Residues Y108 and H109 of the heavy chain of 10D2 (black) make critical contacts to conserved residues V107, W108, and D114 of VP1 (blue), conferring cross-serotype neutralization to PV2 and PV3. Corresponding models are colored based on the provided key (top left), with additional coloring by heteroatom. Starred residues in the VP1 sequence alignment (top right) refer to invariant residues in contact with the 10D2 heavy chain CDR3 loop.

towards PV serotypes 1 and 3, respectively. One of these critical residues is at position 168 of VP1, which is variable for each of the three PV serotypes, being a glutamate in SPV1, a glycine in PV2, and an arginine in SPV3. Whereas 2E1 can neutralize SPV3, binding through electrostatic interactions of E56 (CDR loop 2) with K168 in SPV3. However, the R100 residue (CDR loop 3) in 5E12 could not interact with this same capsid residue due to electrostatic repulsions that would abolish binding or neutralization of SPV3. Additionally, the glycine residue in PV2 has no capability for predicted H-bond formation, leading to a lack of contact and neutralization of PV2 by 5E12, 2E1, and 6B5.

10D2 is the only mAb shown to neutralize multiple serotypes of poliovirus, type 2 and 3, respectively. The 10D2 variable domain shares several contacts on PV capsids with those of 9H2, namely invariant residues at VP1 positions 108 and 114 deep within the canyon and adjacent to the pocket where pocket factor resides. These amino acid identities are invariant in all three PV serotypes, making this site a critical epitope to exploit for biologics development. Importantly, no CDR loop in the 2E1, 5E12, or 6B5 variable domains dips into this canyon region, which may explain the serotype-dependent neutralization seen in these mAbs. Since these residues are conserved for all serotypes of PV, escape mutants for these two positions could possibly lead to reduced PVR binding, release of pocket factor, or defective entry[37,38].

5E12 has 14 shared contacts with PVR and eight contacts with 9H2. Complexes of PV1-PVR and SPV1-9H2 have lost PF[14,23]. Interestingly, the SIPV1-5E12 complex has strong density for pocket factor, equal in magnitude to the virus capsid itself. This presence of the pocket factor could be attributed to the lack of contact in the GH loop at residue 234 of VP1, which interacts with both PVR and 9H2 on type 1 poliovirus. This contact is located on the GH loop adjacent to the pocket, diagonal to the CD loop containing VP1 residues 108 and 114. Inversely, for mAb 10D2, while there are contacts in the CD loop at residues 108 and 114 of VP1, no contact is seen in the GH loop, and pocket factor is retained in the capsid. Contacts of both the CD loop and the GH loop have previously been shown for receptor binding and subsequent pocket factor release for group B enteroviruses such as CVB1 and Echovirus 6[39,40], as well as related group C enteroviruses such as CVA24[41], but to a lesser extent in group A enteroviruses[42,43] or group D enteroviruses[44]. This finding suggests a model in which the release of pocket factor from the poliovirus capsid requires "pinching from both sides" on each of these loops adjacent to the pocket. Without both contacts taking place, the pocket factor will be retained, leading to inhibition of A-particle formation (Fig. 6).

Due to 10D2's ability to neutralize both serotypes 2 and 3 poliovirus, as well as its neutralization targeting an invariant epitope in the PV canyon, 10D2 has potential as a biologic. Due to its conserved epitope, which also appears critical for PVR binding, it is unlikely that escape mutants towards 10D2 would arise, without sacrificing PVR-binding affinity and entry into the host cell[37,38]. With circulating

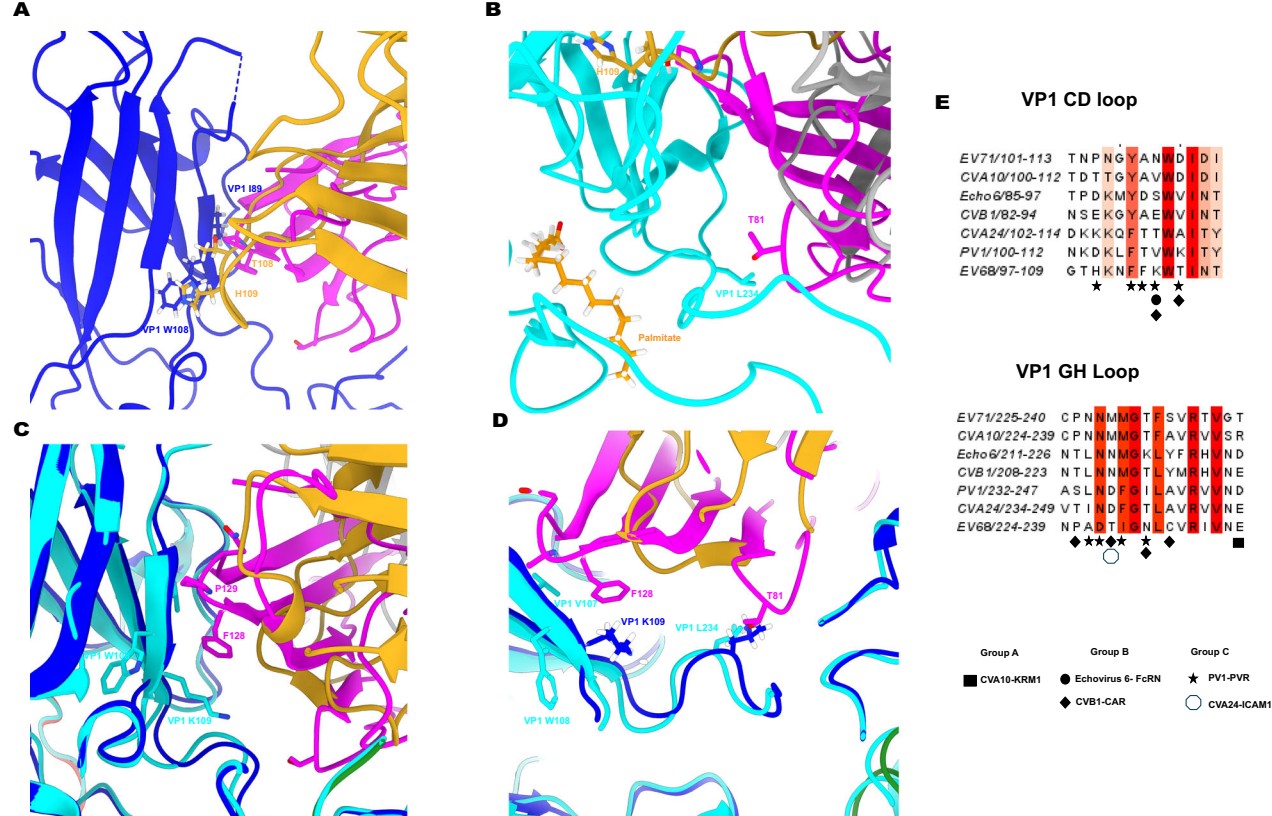

**Fig. 6 | Contacts in C-strand and GH loop of VP1 are critical for pocket factor release.** Capsid proteins VP1 and VP3 of PV–FAb complexes are colored blue and green, respectively, while VP1 of PV–PVR complex is colored cyan., respectively. Heavy and light chains of FAbs are colored goldenrod and dark gray, respectively. Poliovirus receptor is colored magenta. Comparative contacts on the CD loop (**A**) and GH loop (**B**) by PVR and mAb 10D2. Pocket factor is shown as palmitate (orange) with additional coloring by heteroatom. Comparative contacts on the CD loop (**C**) and GH loop (**D**) by PVR and mAb 5E12. **E** Comparison of receptor contacts on VP1 CD loop and VP1 GH loop residues. Receptor contacts are represented using icons as shown.

vaccine-derived poliovirus serotype 2 (cVD-PV2) cases present in several countries globally, even more so than wild-type transmission of poliovirus[45], we believe that 10D2 would serve well to combat these cVD-PV outbreaks and accelerate the eventual eradication of PV transmission.

## Methods

### Mammalian cell culture and viruses

HeLa or Vero cells (ATCC) were infected with poliovirus inoculum at an MOI of 0.1–4 and propagated until >90% CPE was achieved. The virus was then titered by plaque assay as described previously[23]. Sabin serotypes 1 and 3 were kindly provided by Konstantin M. Chumakov, previously at CBER-FDA. After data collection, all wild-type PV2 inoculum, stocks, and associated materials were destroyed in accordance with the US CDC, according to their exact instructions. Sabin-inactivated poliovirus was supplied by the Beijing Institutes of Biological Sciences, China. Virus stock was exchanged against PV buffer (50 mM HEPES, pH 8, 150 mM NaCl, 3 mM MgCl$_2$), as described previously[23].

### Virus propagation and purification

Poliovirus strains were propagated and purified as previously described[23]. Each virus propagation for purification was conducted in a 10 and 2-stack cell factory (VWR). After infection, the cell lysate was frozen and thawed 3 times and centrifuged at 15,000×$g$ for 10 min to pellet cell debris. The debris was resuspended and Dounce homogenized with the addition of 10% NP40 (Thermo Fisher). The homogenization was combined with the supernatant and centrifuged at

15,000×$g$ for 10 min. The resulting supernatant was transferred to a 4 L beaker, PEG 8000 was added to 5%, NaCl was added to a final concentration of 0.5 M, and stirred overnight at 4 °C. The mixture was centrifuged at 15,000×$g$ for 10 min to pellet the PEG-aggregated virus, which was resuspended in a minimal amount of PV Buffer (50 mM HEPES, pH 8, 200 mM NaCl, and 3 mM MgCl$_2$ buffer). 0.5 M MgCl$_2$ was added to 10% of the sample volume, followed by DNAse I (Spectrum) and SDS 10–5% of the sample volume, and incubated at RT for 30 min. Per mL of sample, 0.8 mg trypsin and 0.15 mL of 0.5 M EDTA, pH 9.5, were added and incubated at 37 °C 10 min. N-Lauryl Sarcosine 10% was added to 10% of the sample volume. This sample was then transferred to 50.2 Ti Beckman tubes and 2 mL 30% sucrose in the sample buffer layered beneath it. Tubes were transferred to the Beckman 50.2 Ti rotor for ultracentrifugation at 277,688×$g$ for 2 h at 4 °C with slow braking. The supernatant was discarded, and the viral pellet rehydrated in 1 mL sample buffer at 4 °C overnight. The fully resuspended virus was applied to a continuous 10–40% K-tartrate gradient, balanced in the SW41 rotor, and ultracentrifuged at 222,000×$g$ for 1.5 h at 4 °C with no brake. Resulting virus bands were collected into a syringe by piercing the ultracentrifuge tube with an 18 G needle. Virus was concentrated, and the buffer was exchanged by ultracentrifugation at 277,688×$g$ and resuspension in PV Buffer. Concentration was estimated by spectrophotometry (DeNovix).

### FAb generation from five-fold binding mAbs

All mAbs were provided by the PATH Center for Vaccine Access and Design. Antigen-binding fragments (FAb) were digested using Pierce FAb Micro Preparation Kits (Thermo Fisher, cat# 44685) and purified

using Protein A microcentrifuge columns. Purified FAb was further concentrated using tabletop microcentrifuge columns (Millipore-Sigma).

## Microneutralization assays

Microneutralization assays for five-fold-binding mAbs and FAbs were performed as previously described, with minor modifications[23]. Vero cells grew to ~80% confluence in DMEM, 1% NEAA, and 10% FBS media in 96-well plates. 50 μL 1400 pfu/mL SPV1 or SPV3 virus inoculum was incubated with 50 μL of mAb or FAb (ranging from $2.0*10^{-2}$ to $2.0*10^{-9}$ mg/mL) in DMEM, 1% NEAA for 1 h at 32 °C prior to application to PBS-washed cells. After incubation at 32 °C for 1 h, 50 μL DMEM, 1% NEAA, 5% FBS media was added to each well. Plates were incubated at 32 °C and monitored for CPE by microscopy for 72 h.

## Sample preparation and cryoEM data collection

Poliovirus-FAb complexes were made by incubating FAb and PV together at a ratio of 3 FAb: 1 binding site for 30 min at room temperature. 3.5 μL of complex was applied to glow-discharged 2/1 Cu Quantifoil grids with 3–4 nm continuous carbon coating. Samples were incubated for one minute, blotted, and vitrified using an MK III Vitrobot (Thermo Fisher) operating at 4 °C and 100% humidity. CryoEM datasets for 5E12 and 10D2-PV complexes were collected at 200 kV with a Talos Arctica cryo-TEM (Thermo Fisher) at the Penn State CryoEM facility, and datasets for 2E1 and 6B5-PV complexes were collected using a 300 kV Krios G2i cryo-TEM (Thermo Fisher) at the Hormel Institute. EPU software was used for automated data collection with defocus ranges from −0.5 to −2 μm for each PV-FAb complex. The Talos Arctica was equipped with a Falcon IV direct electron detector at a magnification of ×120,000, yielding a pixel size of 1.3 Å. The Krios G2i was equipped with a Gatan K3 direct electron detector operating at ×59,000 magnification, yielding a pixel size of 1.1 Å (Table S1).

## cryoEM image processing

All cryoEM image processing was performed in CryoSparc[46]. Micrographs were motion corrected, and patch CTF estimation was performed using the "Patch CTF" job. Particles were picked using the "blob picker" and "template picker" jobs, extracted from micrographs, and selected for those with discernible density for the FAb fragment on the capsid and densities corresponding to the intact genome. The best particles were then selected for homogeneous refinement, using an initial model for each complex generated through ab-initio reconstruction and with imposed I1 symmetry (Figs. S9–S12).

## Model building and contact identification

Preliminary models for FAb were generated by submitting the variable domain sequences to SabPred AbodyBuilder2. Initial build of the viral capsid was performed using PDB models 8E8Z, 8E8S, and 8E8X for SIPV1, PV2, and SIPV3-binding mAbs, respectively[23]. Model building and refinement were performed using the ISOLDE plug-in in UCSF ChimeraX[47,48] and Phenix[49], before model validation with MolProbity[50]. Map-to-model fit was assessed using the Q-Score plugin of ChimeraX[51]. FAb contacts on the virus surface were examined in UCSFChimera[34] and defined as residues having atoms separated by <0.4 Å van der Waals radius. Palmitate was modeled as the pocket factor, as it is the most common pocket factor for enteroviruses[52]. Roadmaps defining contacts for FAb, 9H2, and PVR contacts on the virus surface were generated using RIVEM[53]. Figures were generated using ChimeraX. Sequence alignments between PV strains were performed using ClustalOmega[54] and imaged with JalView[55].

## Reporting summary

Further information on research design is available in the Nature Portfolio Reporting Summary linked to this article.

## Data availability

The cryoEM maps of the PV–FAb complexes have been deposited in the EMDB (EMD-70392 (https://www.ebi.ac.uk/emdb/EMD-70392), EMD-70318, EMD-70320 and EMD-70339 for SIPV1-5E12, PV2-10D2, SIPV3-2E1, and SIPV3-6B5, respectively). The coordinates for the PV–FAb complex atomic models have been deposited to the Protein Data Bank (PDB-9OEA, PDB-9OCL, PDB-9OCO, and PDB-9OD3 (for SIPV1-5E12, PV2-10D2, SIPV3-2E1, and SIPV3-6B5, respectively)). Source data for microneutralization assays are provided as a source data file. Source data are provided with this paper.

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

## Acknowledgements

Funding was provided by the Pennsylvania Department of Health Commonwealth Universal Research Enhancement (CURE) funds. Research reported in this publication was supported by the Office of the Director, NIH, under award number S1000D026922-01 (SLH) and NIH 1 RO1 AI10721-01 (SLH) and by the Gates Foundation Grant (INV 010160).

## Author contributions

B.T.W., A.J.C., S.K.D., K.M.C., and S.L.H. conceived the study. B.T.W., A.J.C., J.E.F., and N.M.D. conducted cell culture, virus propagation and purification, biochemistry, and virology. R.D.P., K.M., S.D.D., S.K.D., and K.M.C. provided reagents. S.H.C. and C.M.B. prepared cryoEM samples for data collection and collected data. B.T.W. solved the structures, refined the maps, built the models, and interpreted the results. B.T.W. and S.L.H. interpreted the data and wrote the manuscript.

## Competing interests

The authors declare no competing interests.
