## [Transparent Peer Review file · Nature Communications]

Neutralizing human monoclonal antibodies to poliovirus map to the receptor binding site

Corresponding Author: Dr Susan Hafenstein

Version 0:

Reviewer comments:

Reviewer #1

(Remarks to the Author)

Poliovirus eradication is increasingly uncertain, spurring important research in the field. Building upon their recent publication (Nat Comm 2023), the authors used cryo-EM to solve high resolution structures of four poliovirus-Fab complexes. The Fabs, obtained from neutralizing mAbs 10D2, 5E12, 2E1 and 6B5, bound to icosahedral poliovirus particles in a canyon around the 5-fold axis of symmetry. Contacts with serotype-specific capsid residues were identified for three of the Fabs while a cross-neutralizing mAb (10D2) contacts conserved residues located deep in the capsid canyon, reinforcing an outcome reported before (Nat Comm 2023). The Fabs bind to overlapping regions of the canyon required for binding of the poliovirus receptor, suggesting a mechanism of neutralization that prevents receptor binding and entry. Intriguingly, unlike binding of the poliovirus receptor, the Fabs do not provoke large conformational changes in capsid structures nor induce release of pocket factor. Altogether, this study highlights how Fab-poliovirus protein-protein interactions that neutralize poliovirus particles are similar, yet distinct from receptor-poliovirus protein-protein interactions that trigger various aspects of virus binding and entry (pocket factor release, A-particle formation and genome release).

The authors conclude that serotype-specific mAbs contact serotype-specific residues of capsid proteins whereas cross-neutralizing mAbs contact more conserved residues deep in the capsid canyon.

Critique: The current investigation is consistent with, and builds upon, a large body of work in the field. The data for 10D2 cross-neutralizing antibodies reported herein reinforce those previously reported for 9H2 (Nat Comm 2023): cross-neutralizing antibodies contact conserved residues deep in the capsid canyon. Altogether, this study highlights and compares Fab-poliovirus protein-protein interactions that neutralize poliovirus particles with receptor-poliovirus protein-protein interactions that trigger various aspects of virus binding and entry (pocket factor release, A-particle formation and genome release). Unlike binding of the poliovirus receptor, the Fabs do not provoke large conformational changes in capsid structures nor induce release of pocket factor.

The data are of high quality, supporting the authors important conclusions.

I have only minor points for the authors' consideration.

Minor points

1. Line 102. Awkward phrase...biologicals towards poliovirus.
2. Line 164. Spell pocket factor.
3. Discussion, lines 262-273. Is there a logical reason why distinct poliovirus epitopes are reported for murine and human antibodies?

Reviewer #2

(Remarks to the Author)

The main new findings are the role of residue 100 in the restriction of some antibodies to specific serotypes, and a proposed role of residue 234 in pocket factor release. The structural work is new but does not bring the type of conceptual or mechanistic insights expected in Nature Communications. This is all the more important since the "real-life" impact of this

work is not directly evident. A useful antibody exists (9H2) and there is no real evidence that the delivery of antibodies is a viable solution in the countries affected by endemic poliovirus 1 or outbreaks of vaccine-derived poliovirus outbreaks. These are often remote locations, war zone or under-resourced regions where injection with purified antibody is not an option.

Fundamentally, the mechanism of antibody-induced release of the pocket factor would be interesting, but this study does not provide data beyond static structures (indeed, these antibodies don't induce the release of the pocket factor).

As for the form, the design and quality of the figures does not support the interpretations. They are difficult to read at best and do not always illustrate the point made in the text:

Fig. 1 belongs to the supplementary material (at least for panels B and C).

Fig. 2 is very hard to read. The black colour and choice of shadows/rendering makes it uninterpretable. Zoomed insets on what is to take away from the figure would help.

Fig. 4: again, poor rendering choice. White-on-white hydrogen atoms are almost invisible.

Fig. 6: Is it not evident at all from Fig. 6 that residue 234 is "critical for pocket factor release". We just see a couple of loops that swing in/out but don't have any idea of pocket size, opening, surface properties.

Technically, I'm surprised that datasets with 10 times more particles and a better microscope/detector setup yields slightly worse resolution to the Glacios one (~2.9Å). They should be much better (and the local resolution map support that the global resolution is better than stated). Whether it is an error in reporting or in data processing, this is concerning for a straightforward Cryosparc workflow.

Detailed comments:

l. 23-24. "vaccine-derived poliovirus (VDPV) has exacerbated the current situation": In the current anti-vaccine climate, I would be cautious with the way these points are articulated.

l. 35-36: How are these going to be delivered in practice in area of concern? Is serotype-specificity important?

l. 44, "most severe clinical outcome of acute flaccid paralysis": no, the most severe clinical outcome is death.

l. 82, If this one antibody (9H2) is known and broad-spectrum, is there still a need for new mAbs?

l. 244, monovalent binding: If it's true that the binding of intact IgG is monovalent, this should be measurable in neutralisation assays with similar EC50s between Fab and IgG when corrected for concentration. Is this the case? The figure supporting the monovalent binding, tangential orientation etc. should be presented in supplementary material.

Version 1:

Reviewer comments:

Reviewer #1

(Remarks to the Author)

Waddey and colleagues report that neutralizing human monoclonal antibodies to poliovirus map to the receptor binding site. The overlapping footprints and specific protein-protein interactions of both mAbs and poliovirus receptor are clearly highlighted in the revised Figures and Tables. The cryo-EM structures of poliovirus-FAb complexes are of sufficient resolution to support the author's important conclusions.

As summarized in their response to the previous critiques, the manuscript was revised to improve some aspects of data presentation and to clarify cryo-EM data collection and processing methods. The authors clarified the factors that impact the resolution of poliovirus-FAb complexes. Altogether, revisions in the manuscript were responsive to the previous critiques, improving the clarity and impact of the study.

Theoretically appealing conceptual frameworks. 1) FAb-poliovirus protein-protein interactions that neutralize poliovirus particles are similar, yet distinct from receptor-poliovirus protein-protein interactions that trigger various aspects of virus binding and entry (pocket factor release, A-particle formation and genome release). Intriguingly, unlike binding of the poliovirus receptor, the FAbs do not provoke large conformational changes in capsid structures nor induce release of pocket factor. 2) Serotype-specific mAbs contact serotype-specific residues of capsid proteins whereas cross-neutralizing mAbs contact more conserved residues deep in the capsid canyon. These are theoretically appealing conceptual frameworks to inspire ongoing work in the field.

Reviewer #2

(Remarks to the Author)

The authors have not rebutted the main points from my previous review (1st two paragraphs) and focused on my comments on the figures. From my reading, both reviewers agree that given the previous publication (Nature Communications, 2023) and the absence of a novel mechanism, the work does not bring the type of conceptual advances expected for publication in Nature Communications. Nothing in the revised submission counters these initial views.

My concern about the quality of the structural work remains. With 237k and 450k icosahedral particles, the resolution limits should be better. More advanced processing should have been explored.

POINT-BY-POINT RESPONSE

Review round: 1

Manuscript ID: NCOMMS-25-54606-T

Title: "Neutralizing human monoclonal antibodies to poliovirus map to the receptor binding site"

REVIEWER COMMENTS

Reviewer #1 (Remarks to the Author):

Poliovirus eradication is increasingly uncertain, spurring important research in the field. Building upon their recent publication (Nat Comm 2023), the authors used cryo-EM to solve high resolution structures of four poliovirus-Fab complexes. The Fabs, obtained from neutralizing mAbs 10D2, 5E12, 2E1 and 6B5, bound to icosahedral poliovirus particles in a canyon around the 5-fold axis of symmetry. Contacts with serotype-specific capsid residues were identified for three of the Fabs while a cross-neutralizing mAb (10D2) contacts conserved residues located deep in the capsid canyon, reinforcing an outcome reported before (Nat Comm 2023). The Fabs bind to overlapping regions of the canyon required for binding of the poliovirus receptor, suggesting a mechanism of neutralization that prevents receptor binding and entry. Intriguingly, unlike binding of the poliovirus receptor, the Fabs do not provoke large conformational changes in capsid structures nor induce release of pocket factor. Altogether, this study highlights how Fab-poliovirus protein-protein interactions that neutralize poliovirus particles are similar, yet distinct from receptor-poliovirus protein-protein interactions that trigger various aspects of virus binding and entry (pocket factor release, A-particle formation and genome release).

The authors conclude that serotype-specific mAbs contact serotype-specific residues of capsid proteins whereas cross-neutralizing mAbs contact more conserved residues deep in the capsid canyon.

Critique: The current investigation is consistent with, and builds upon, a large body of work in the field. The data for 10D2 cross-neutralizing antibodies reported herein reinforce those previously reported for 9H2 (Nat Comm 2023): cross-neutralizing antibodies contact conserved residues deep in the capsid canyon. Altogether, this study highlights and compares Fab-poliovirus protein-protein interactions that neutralize poliovirus particles with receptor-poliovirus protein-protein interactions that trigger various aspects of virus binding and entry (pocket factor release, A-particle formation and genome release). Unlike binding of the poliovirus receptor, the Fabs do not provoke large conformational changes in capsid structures nor induce release of pocket factor.

The data are of high quality, supporting the authors important conclusions.

We appreciate the comments by Reviewer 1.

I have only minor points for the authors' consideration.

Minor points

1. Line 102. Awkward phrase...biologicals towards poliovirus.

The text was changed to be less awkward.

2. Line 164. Spell pocket factor.

This has been corrected.

3. Discussion, lines 262-273. Is there a logical reason why distinct poliovirus epitopes are reported for murine and human antibodies?

Citations have been added to this passage. Historically, epitope mapping of poliovirus was done with murine antibodies and escape mutants (25-27). Chumakov (24) reported the first epitope mapped with a chimpanzee-human chimeric antibody, which mapped to a different location on the capsid from any of the work done with murine antibodies. Our lab reported the first epitope mapped to the virus capsid with a human monoclonal, 9H2 (23). Here we report how four more human monoclonal antibodies recognize the capsid. We have been surprised throughout these studies that there is little to no overlap between human and murine antibodies on the capsid.

Reviewer #2 (Remarks to the Author):

The main new findings are the role of residue 100 in the restriction of some antibodies to specific serotypes, and a proposed role of residue 234 in pocket factor release. The structural work is new but does not bring the type of conceptual or mechanistic insights expected in Nature Communications. This is all the more important since the “real-life” impact of this work is not directly evident. A useful antibody exists (9H2) and there is no real evidence that the delivery of antibodies is a viable solution in the countries affected by endemic poliovirus 1 or outbreaks of vaccine-derived poliovirus outbreaks. These are often remote locations, war zone or under-resourced regions where injection with purified antibody is not an option.

Fundamentally, the mechanism of antibody-induced release of the pocket factor would be interesting, but this study does not provide data beyond static structures (indeed, these antibodies don't induce the release of the pocket factor).

As for the form, the design and quality of the figures does not support the interpretations. They are difficult to read at best and do not always illustrate the point made in the text:

Fig. 1 belongs to the supplementary material (at least for panels B and C).

Panels C showing fit to map have been moved to Supplemental leaving only panels A and B showing surface-rendered and central sections of

maps. For complexes, illustrating the central section is necessary to show the quality of the map and the density magnitude of the bound protein relative to that of the capsid.

Fig. 2 is very hard to read. The black colour and choice of shadows/rendering makes it uninterpretable. Zoomed insets on what is to take away from the figure would help.

The figure has been reworked with changes in color and the addition of zoomed insets to aid interpretation.

Fig. 4: again, poor rendering choice. White-on-white hydrogen atoms are almost invisible.

The figure has been reworked with changes in color to aid interpretation.

Fig. 6: Is it not evident at all from Fig. 6 that residue 234 is "critical for pocket factor release". We just see a couple of loops that swing in/out but don't have any idea of pocket size, opening, surface properties.

To address the Reviewer's concern, we have reworked the figure, edited and added text, and included additional analysis. These changes add clarification to the presentation of residues critical for pocket factor release.

Technically, I'm surprised that datasets with 10 times more particles and a better microscope/detector setup yields slightly worse resolution to the Glacios one (~2.9Å). They should be much better (and the local resolution map support that the global resolution is better than stated). Whether it is an error in reporting or in data processing, this is concerning for a straightforward Cryosparc workflow.

We have added a description of the maps reporting the resolution as a range, which as the Reviewer points out, is an important description for a reconstruction of a complex where the virus and the site of interaction are often a better resolution than the apical tip of the bound protein.

Workflow figures have been added to supplemental.

Two data sets (5E12 and 10D2) had about 20,000 particles each collected on the Glacios and attained resolutions of 2.84 and 3.65, respectively. Two data sets (2E1 and 6B5) consisted of 237k and 450k particles each were collected on the Krios G2 and attained resolutions of 2.95 and 2.94, respectively (Table S1). Both microscopes in question have been benchmarked as capable of producing maps at 1.9 Å resolution. It is not commonly recognized that a 200 kV Glacios or Arctica is capable of 1.9 Å resolution.

Although there is a correlation between particle number and resolution, there are multiple other factors controlling final resolution such as ice thickness and protein flexibility. For both of the data sets collected on the Krios, we noticed areas of thicker ice.

Detailed comments:

I. 23-24. "vaccine-derived poliovirus (VDPV) has exacerbated the current situation": In the current anti-vaccine climate, I would be cautious with the way these points are articulated.

We agree with this caution and have edited the text. The passage in Abstract was deleted entirely.

I. 35-36: How are these going to be delivered in practice in area of concern? Is serotype-specificity important?

There are currently no approved and licensed antivirals or therapeutics such as monoclonals for treatment of iVDPV subjects globally, the treatment of these subjects will be coordinated with WHO GPEI (Global Polio Eradication Initiative) to treat the handful of iVDPV subjects who are chronic poliovirus shedders and specificity of the antibody in this treatment option to such subjects will help to achieve the two main goals of polio eradication (wildtype virus eradication and VDPVs including iVDPV eradication) (10).

I. 44, "most severe clinical outcome of acute flaccid paralysis": no, the most severe clinical outcome is death.

The text has been corrected.

I. 82, If this one antibody (9H2) is known and broad-spectrum, is there still a need for new mAbs?

A critical challenge to achieving complete eradication of poliovirus circulation will be shedding of live vaccine-derived poliovirus strains by individuals with primary immunodeficiencies (PID). Because live poliovirus shedding can persist for years in these individuals, chronic excretion presents a long-term risk of seeding ongoing VDPV circulation and potentially disease outbreaks (10). A therapeutic intervention which can clear poliovirus infection from a substantial fraction of this population could mitigate this risk. Intervention options including use of individual broad spectrum or in combination with type specific antibodies will be very helpful options.

I. 244, monovalent binding: If it's true that the binding of intact IgG is monovalent,

this should be measurable in neutralisation assays with similar EC50s between Fab and IgG when corrected for concentration. Is this the case? The figure supporting the monovalent binding, tangential orientation etc. should be presented in supplementary material.

The measurements to assess whether bivalent binding is possible were established in seminal papers by Elizabeth Hewat and Dieter Blaas (35,36), which are cited in the manuscript. In the four complex structures solved and presented here, none of the Fabs are bound with a geometry consistent with bivalent binding. Specifically (lines 243-246), Fabs are not bound in an orientation relative to the capsid or to the symmetry related neighboring fab that would allow bivalent binding of a Mab. The text has been edited to clarify, with care taken to avoid overstating the analysis.